# Sparse Backpropagation for MoE Training

## Abstract

One defining characteristic of Mixture-of-Expert (MoE) models is their capacity for conducting sparse computation via expert routing, leading to remarkable scalability. However, backpropagation, the cornerstone of deep learning, requires dense computation, thereby posting challenges in MoE gradient computations. Here, we introduce SparseMixer, a scalable gradient estimator that bridges the gap between backpropagation and sparse expert routing. Unlike typical MoE training which strategically neglects certain gradient terms for the sake of sparse computation and scalability, SparseMixer provides scalable gradient approximations for these terms, enabling reliable gradient estimation in MoE training. Grounded in a numerical ordinary differential equations (ODE) framework, SparseMixer harnesses the mid-point method, a second-order ODE solver, to deliver precise gradient approximations with negligible computational overhead. Applying SparseMixer to Switch Transformer on both pre-training and machine translation tasks, SparseMixer showcases considerable performance gain, accelerating training convergence by up to 2 times[1].

## 1 Introduction

The significant success of large-scale pre-training across various applications has underscored the imperative need for scalable models that are economically feasible (Chowdhery et al., 2022; OpenAI, 2023; Touvron et al., 2023). Recent advances in sparsely activated networks, prominently known as Mixture-of-Experts (MoE), have attracted widespread interest (Shazeer et al., 2017; Lepikhin et al., 2020; Fedus et al., 2021; Riquelme et al., 2021; Mustafa et al., 2022). Unlike traditional networks that densely activate all modules for all input, MoE selectively activates parts of modules to specific inputs through a process called expert routing, leading to notable efficiency enhancements.

However, such efficiency gain comes at a cost: gradient estimation in MoE becomes challenging due to expert routing. Specifically, the routing function, being discrete in nature, produces non-differentiable outputs. Meanwhile, backpropagation, the cornerstone of deep learning, relies on the Chain rule, making it exclusively compatible with differentiable functions (Rosenblatt, 1957; Bengio et al., 2013), and cannot be directly applied for gradient computation of expert routing.

Numerous methods have emerged to bridge discrete and back-propagation, and most of them are based on Straight-Through (ST) (Rosenblatt, 1957; Bengio et al., 2013; Jang et al., 2017; Liu et al., 2023). Unfortunately, all existing ST estimators are incompatible with MoE, since they require activating all experts for gradient computing, thereby eliminating all the efficiency improvements of MoE. Consequently, typical MoE training strategically neglects the gradient computation for routing, trading certain training signals for sparse computation. Despite the scalability brought by sparse computation, this trade-off may result in slow convergence and improperly trained models.

Our solution to this quandary is SparseMixer—a novel approach designed to reconcile the divide between sparse MoE routing and backpropagation. Drawing inspiration from numerical methods for ordinary differential equations (ODE), SparseMixer provides reliable gradient approximation for expert routing, even when only a subset of experts are activated. Moreover, to furnish accurate gradient approximations with negligible computation overheads, we integrate the mid-point method, a second-order numerical ODE solver, which matches the Taylor expansion of the gradient to the second order without requiring the Hessian matrix or other second-order derivatives.

---

[1]Reproducibility: Implementations are available at the supplemental materials and will be released publicly.

We apply SparseMixer to Switch Transformer on both pretraining and neural machine translation. SparseMixer not only accelerates training convergence by up to two times but also facilitates MoE with properly trained expert routing. Remarkably, while Switch Transformer underperforms the dense model in all three pretraining settings, incorporating SparseMixer as the gradient estimator allows the resulting MoE models to consistently outperform the dense model.

## 2 RELATED WORK AND PRELIMINARY

**Mixture-of-Expert for Transformer.** The idea of Mixture-of-Expert models originates from Jacobs et al. (1991) and Jordan & Jacobs (1994), which integrates many separate networks together and uses each to handle a separate subset of training cases. Recently, many attempts have been made to leverage this idea for scaling large language models (Shazeer et al., 2017; Lepikhin et al., 2020; Lewis et al., 2021; Fedus et al., 2021).

To keep things straightforward, we will focus on a simplified setting of the switch Transformer layer (Fedus et al., 2021), and the resulting algorithm can be easily extended to other MoE designs. Considering a set of N experts, $\{f_i(\boldsymbol{x})\}_{i=1}^N$, the gate value of expert $i$ is computed with the softmax function as $\boldsymbol{\pi}_i = \text{softmax}(\boldsymbol{\theta})_i = \frac{\exp(\boldsymbol{\theta}_i)}{\sum_{j=1}^n \exp(\boldsymbol{\theta}_j)}$, where $\boldsymbol{\theta} = W_r \cdot \boldsymbol{x}$. For $i \in [1, \cdots, N]$, we mark its one-hot representation as $\boldsymbol{I}_i \in \mathcal{R}^{N \times 1}$, whose element equals 1 if it is the $i$-th element or equals 0 otherwise. Let $\boldsymbol{D}$ be a discrete random variable and $\boldsymbol{D} \in \{\boldsymbol{I}_1, \cdots, \boldsymbol{I}_N\}$. Then, the final output of this MoE layer is $\boldsymbol{y} = \boldsymbol{\pi}_{\boldsymbol{D}} f_{\boldsymbol{D}}(\boldsymbol{x})$. Note that $\boldsymbol{D}$ is sampled as $\boldsymbol{D} \sim \boldsymbol{\pi}$ during training, and is computed as $\boldsymbol{D} \leftarrow \arg\max_{\boldsymbol{I}_i} \boldsymbol{\pi}_{\boldsymbol{I}_i}$ during inference. Marking other parts of the neural network as a differentiable function $g : \mathcal{R}^n \to \mathcal{R}$, we aim to minimize:

$$\min_{W_r} \mathcal{L}(W_r), \text{ where } \mathcal{L}(W_r) = E_{\boldsymbol{D} \sim \text{softmax}(W_r \boldsymbol{x})}[g(\boldsymbol{\pi}_{\boldsymbol{D}} f_{\boldsymbol{D}}(\boldsymbol{x}))] = \sum_{\boldsymbol{D}} \boldsymbol{\pi}_{\boldsymbol{D}} \cdot g(\boldsymbol{\pi}_{\boldsymbol{D}} f_{\boldsymbol{D}}(\boldsymbol{x})). \quad (1)$$

First, we focus our discussions on this simplified MoE model. In Section 3.3, we will discuss its difference with the Switch Transformer and necessary adaptations.

**Gradient Computation for Expert Routing.** For simplicity, we mark $\frac{\partial \mathcal{L}(W_r)}{\partial W_r}$ as $\nabla_0 + \nabla_1$:

$$\frac{\partial \mathcal{L}}{\partial W_r} := \nabla_0 + \nabla_1, \text{where } \nabla_0 = \sum_{\boldsymbol{I}_i} g(\boldsymbol{\pi}_{\boldsymbol{I}_i} f_{\boldsymbol{I}_i}(\boldsymbol{x})) \frac{\partial \boldsymbol{\pi}_{\boldsymbol{I}_i}}{\partial W_r} \text{ and } \nabla_1 = \sum_{\boldsymbol{I}_i} \boldsymbol{\pi}_{\boldsymbol{I}_i} \frac{\partial g(\boldsymbol{\pi}_{\boldsymbol{I}_i} f_{\boldsymbol{I}_i}(\boldsymbol{x}))}{\partial W_r}. \quad (2)$$

It is easy to notice that $\nabla_1$ can be computed reliably via backpropagation. $\nabla_0$, however, is hard to reliably estimate in typical MoE training practice. In this study, we focus our discussions on $\nabla_0$.

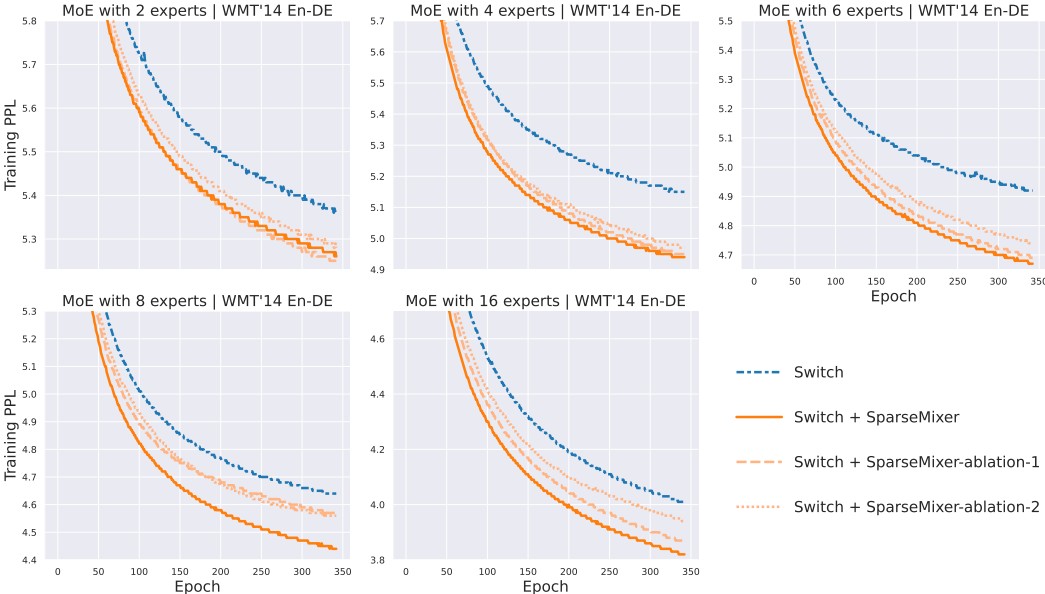

Figure 1: Training curves of Switch Transformer on WMT'14 En-De.

REINFORCE (Williams, 1992) is unbiased (i.e., $E[\nabla_{\text{REINFORCE}}] = \nabla_0$) and only requires the distribution of the discrete variable to be differentiable (i.e., no backpropagation through $g$):

$$\nabla_{\text{REINFORCE}} := g(\boldsymbol{\pi_D} f_{\boldsymbol{D}}(\boldsymbol{x})) \frac{\partial \log \boldsymbol{\pi_D}}{\partial W_r}. \tag{3}$$

Despite the $\nabla_{\text{REINFORCE}}$ estimator being unbiased, it tends to have prohibitively high variance, especially for networks that have other sources of randomness (i.e., dropout or other independent random variables). Recently, attempts have been made to reduce the variance of REINFORCE (Gu et al., 2016; Tucker et al., 2017; Grathwohl et al., 2018; Shi et al., 2022). Still, it has been found that the REINFORCE-style estimators fail to work well in MoE training (Kool et al., 2021).

**Straight-Through.** Despite $\nabla_{\text{REINFORCE}}$ being unbiased, it treats the remaining network ($g$) as a black-box and only leverages the zero-order information of $g$. In practice, a popular family of estimators, Straight-Through (ST), leveraging the first-order information of $g$ (note that $g$ is a scalar and $g'$ is a vector), has been shown to achieve a better performance in more complicated settings (Liu et al., 2023). ST computes the backpropagation "through" a surrogate that treats the non-differentiable function (e.g., the sampling of $\boldsymbol{D}$) as an identity function (Rosenblatt, 1957; Bengio et al., 2013; Jang et al., 2017; Liu et al., 2023). In our MoE setting, ST treats the sampling of $\boldsymbol{D}$ as an identity function and estimates the gradient as:

$$\widehat{\nabla}_{\text{ST}} := \frac{\partial g(\boldsymbol{\pi_D} f_{\boldsymbol{D}}(\boldsymbol{x}))}{\partial \boldsymbol{\pi_D} f_{\boldsymbol{D}}(\boldsymbol{x})} \frac{\partial \sum_i \boldsymbol{D}_i \boldsymbol{\pi_{I_i}} f_{\boldsymbol{I_i}}(\boldsymbol{x})}{\partial \boldsymbol{D}} \frac{\partial \boldsymbol{\pi_D}}{\partial W_r}. \tag{4}$$

An alternative strategy is to conduct the concrete random variable relaxation (Maddison et al., 2014; Jang et al., 2017). It is observed that the sampling of $\boldsymbol{D}$ can be reparameterized using Gumbel random variables at the zero-temperature limit of the tempered softmax (Gumbel, 1954):

$$\boldsymbol{D} = \lim_{\tau \to 0} \boldsymbol{S}_\tau, \text{ where } \boldsymbol{S}_\tau = \text{softmax}_\tau(\boldsymbol{\theta} + \boldsymbol{G}), \boldsymbol{G}_i \text{ are i.i.d., and } \boldsymbol{G}_i \sim \text{Gumbel}(0, 1).$$

Straight-Through Gumbel-Softmax (STGS) treats the zero-temperature limit as identity function during the backpropagation:

$$\widehat{\nabla}_{\text{STGS}} := \frac{\partial g(\boldsymbol{\pi_D} f_{\boldsymbol{D}}(\boldsymbol{x}))}{\partial \boldsymbol{\pi_D} f_{\boldsymbol{D}}(\boldsymbol{x})} \frac{\partial \sum_i \boldsymbol{S}_{\tau,i} \boldsymbol{\pi_{I_i}} f_{\boldsymbol{I_i}}(\boldsymbol{x})}{\partial \boldsymbol{S}_\tau} \frac{\partial \boldsymbol{S}_\tau}{\partial W_r}. \tag{5}$$

Although $E[\widehat{\nabla}_{\text{ST}}]$ has been formally established as a first-order approximation of $\nabla_0$ (Liu et al., 2023), applying ST estimators necessitates the need for computing $f_i(\boldsymbol{x})$ for all $i \in \{\boldsymbol{I_1}, \cdots, \boldsymbol{I_N}\}$, i.e., the outputs from all experts. For example, in Equation 4, we have $\frac{\partial \sum_i \boldsymbol{D}_i \boldsymbol{\pi_{I_i}} f_{\boldsymbol{I_i}}(\boldsymbol{x})}{\partial \boldsymbol{D}} = \text{diag}(\sum_i \boldsymbol{D}_i \boldsymbol{\pi_{I_i}} f_{\boldsymbol{I_i}}(\boldsymbol{x}))$, which involves the computation of $\{f_{\boldsymbol{I_1}}(\boldsymbol{x}), \cdots, f_{\boldsymbol{I_N}}(\boldsymbol{x})\}$. Essentially, computing all $f_{\boldsymbol{I_i}}$ turns MoE into a densely activated network. Thus, using ST-style estimators undermines the sparse computation, fundamentally obstructing the scaling of MoE models.

## 3 Scalable Gradient Approximation

As discussed in Section 2, although ST estimators bridged discrete variables and backpropagation, they require the network to be densely activated. Here, we first discuss the intrinsic limitation of ST estimators. Then, we go beyond ST and bridge sparse expert routing and backpropagation. Finally, we revisit the current practice of MoE training and discuss the difference between the Switch Transformer and the simplified setting (as presented in Section 2).

### 3.1 Why Existing ST Estimators Are Not Scalable?

Liu et al. (2023) formally establishes that $E[\widehat{\nabla}_{\text{ST}}]$ is a first-order approximation of $\nabla_0$ (note that $\nabla_0$ is defined in Equation 2). Since $\boldsymbol{D}_i = 1 \iff \boldsymbol{D} = \boldsymbol{I}_i$, we can reparameterizing $\boldsymbol{\pi_D} f_{\boldsymbol{D}}$ as $h(\boldsymbol{D}) = \sum_i \boldsymbol{D}_i \boldsymbol{\pi_{I_i}} f_{\boldsymbol{I_i}}$. Then, we have[2]:

$$\nabla_0 = \sum_{\boldsymbol{I_i}} (h(\boldsymbol{I}_i) - E[h]) \frac{\partial \boldsymbol{\pi_{I_i}}}{\partial W_r} = \sum_{\boldsymbol{I_i}} \sum_{\boldsymbol{I_j}} \boldsymbol{\pi_{I_j}} (h(\boldsymbol{I}_i) - h(\boldsymbol{I}_j)) \frac{\partial \boldsymbol{\pi_{I_i}}}{\partial W_r}. \tag{6}$$

---

[2]Commonly referred to as baseline subtraction. Note $\sum_i E[g] \frac{\partial \boldsymbol{\pi_{I_i}}}{\partial W_r} = E[g] \frac{\partial \sum_{I_i} \boldsymbol{\pi_{I_i}}}{\partial W_r} = E[g] \frac{\partial \mathbf{1}}{\partial W_r} = 0$.

Specifically, approximating $g(\pi_{\boldsymbol{I}_i} f_{\boldsymbol{I}_i}) - g(\pi_{\boldsymbol{I}_j} f_{\boldsymbol{I}_j})$ as $g'(\pi_{\boldsymbol{I}_j} f_{\boldsymbol{I}_j}) \cdot (\pi_{\boldsymbol{I}_i} f_{\boldsymbol{I}_i} - \pi_{\boldsymbol{I}_j} f_{\boldsymbol{I}_j})$, the resulting gradient approximation will have the same form as $E[\widehat{\nabla}_{\text{ST}}]$ (Liu et al., 2023). In numerical analyses, this approximation is known as the forward Euler method (briefly introduced in Appendix A), which has first-order accuracy. Liu et al. (2023) also explored higher-order ODE solvers to better approximate $g(\pi_{\boldsymbol{I}_i} f_{\boldsymbol{I}_i}) - g(\pi_{\boldsymbol{I}_j} f_{\boldsymbol{I}_j})$. However, all these approximations involve both activated experts (i.e., $f_{\boldsymbol{I}_i}$) and unactivated experts (i.e., $f_{\boldsymbol{I}_j}$), thus contradicting scalability. In order words, although those ST estimators bridge discrete and backpropagation, their computations are dense instead of sparse.

## 3.2 Expert Routing Gradient Approximation: Backpropagation Made Sparse

To bridge the gap between sparse MoE routing and back-propagation, we need to approximate $\nabla_0$ without requiring outputs from all experts. In our study, we move beyond ST and present a novel framework to bridge backpropagation and sparse expert routing.

Here, we start by introducing the most simple gradient estimator, i.e., $\widehat{\nabla}_{\text{SparseMixer -1st}}$, where

$$\widehat{\nabla}_{\text{SparseMixer -1st}} := \frac{\partial g(\pi_{\boldsymbol{D}} f_{\boldsymbol{D}}(\boldsymbol{x}))}{\partial W_r}.$$

Similar to $E[\widehat{\nabla}_{\text{ST}}]$, $E[\widehat{\nabla}_{\text{SparseMixer -1st}}]$ is a first-order approximation of $\nabla_0$. To demonstrate this, we take an alternative approach to rewrite $\nabla_0$:

$$\nabla_0 = \sum_{\boldsymbol{I}_i} (g(\pi_{\boldsymbol{I}_i} f_{\boldsymbol{I}_i}) - g(\boldsymbol{0})) \frac{\partial \pi_{\boldsymbol{I}_i}}{\partial W_r}. \tag{7}$$

Note that $g(\boldsymbol{0})$ is only used as a vehicle for derivations. It is worth mentioning that, unlike the baseline used in Equation 4 (i.e., $E[h]$, which has been shown to be the optimal control variate in Weaver & Tao, 2001), it is not a suitable to use $g(\boldsymbol{0})$ as the control variate for policy gradient.

Adopting the Euler method to Equation 7, we estimate $g(\pi_{\boldsymbol{I}_i} f_{\boldsymbol{I}_i}) - g(\boldsymbol{0})$ as $g'(\pi_{\boldsymbol{I}_i} f_{\boldsymbol{I}_i}) \cdot \pi_{\boldsymbol{I}_i} f_{\boldsymbol{I}_i}$. Comparing to the first-order approximation of Equation 6, this first-order approximation only requires the output of one expert. Then, it is easy to note:

$$\nabla_0 \overset{\text{forward Euler}}{\approx} \sum_{\boldsymbol{I}_i} g'(\pi_{\boldsymbol{I}_i} f_{\boldsymbol{I}_i}) \cdot \pi_{\boldsymbol{I}_i} f_{\boldsymbol{I}_i} \cdot \frac{\partial \pi_{\boldsymbol{I}_i}}{\partial W_r} = E_{\boldsymbol{D} \sim \boldsymbol{\pi}} [\frac{\partial g(\pi_{\boldsymbol{D}} f_{\boldsymbol{D}}(\boldsymbol{x}))}{\partial W_r}] = E[\widehat{\nabla}_{\text{SparseMixer -1st}}].$$

Note that, same with $\widehat{\nabla}_{\text{ST}}$, $\widehat{\nabla}_{\text{SparseMixer -1st}}$ adopts the forward Euler method and achieves first-order accuracy. Meanwhile, $\widehat{\nabla}_{\text{SparseMixer -1st}}$ only requires the output of one expert thus not sacrificing scalability, while $\widehat{\nabla}_{\text{ST}}$, as in Equation 4, requires the output of all experts.

## 3.3 Understand Current MoE Training Practice in Simplified Setting

Besides providing sound gradient approximation with negligible computation overheads, our study also sheds insights into the underlying mechanism of the current MoE training practice. In this section, we introduce the current MoE training practice with our simplied setting. Then, in Section 3.4, we further discuss MoE training in the realistic Switch Transformer setting.

**Current MoE Training Practice.** Due to all the challenge discussed in Section 3.1, the current MoE training practice trades certain training signals for scalability. Specifically, $\nabla_0$ is strategically neglected in gradient computation (the value of $\nabla_0$ is set to 0), and only $\nabla_1$ is used for model training (Fedus et al., 2021). Despite the success of such practice, it remains unclear on the impact of neglecting $\nabla_0$, how to conduct training with only part of the gradient, and whether gradient descent is still effective after neglecting $\nabla_0$.

**Underlying Mechanism of Current MoE Training Practice.** Comparing Equation 7 and Equation 2, we can observe that $\widehat{\nabla}_{\text{SparseMixer -1st}}$ has the same form with $\nabla_1$, which implies:

$$\nabla \overset{\text{forward Euler}}{\approx} 2 \cdot \nabla_1.$$

Therefore, in our simplified setting, directly dropping the $\nabla_0$ can be viewed as down-scaling $\nabla$ by 0.5. Since typical training practice employs adaptive optimizers for model training, whose update rule is invariant to constant gradient scaling, our observation here provides a natural explanation on the effectiveness of current MoE training.

### 3.4 FROM SIMPLIFIED SETTING TO SWITCH TRANSFORMER: EXPERT SAMPLING

As mentioned in Section 2, our modeling of MoE is a simplified Switch Transformer. Here, we first discuss the difference between our simplified setting and Switch Transformer, and then move to necessary modifications to apply SparseMixer to Switch Transformer.

**Difference between Simplified Setting and Switch Transformer.** The difference between our simplified setting and switch Transformer is the sampling of $D$. Specifically, in our simplified setting, we assume $D$ is sampled from $\pi$; in Switch Transformer, $D$ is sampled as Equation 8 instead.

$$D = \arg\max_{I_i}(\boldsymbol{\theta}_{I_i} \cdot u_{I_i}), \text{ where } u_{I_i} \overset{\text{iid}}{\sim} \text{Uniform}(1-r, 1+r). \tag{8}$$

With this sampling strategy, $\nabla_1$ is no longer a first-order approximation to $\nabla_0$ (it can be viewed as conducting importance sampling without applying the likelihood ratio).

**An Important Property of Expert Sampling in Switch Transformer.** To obtain sound gradient estimation with a strong performance, it is necessary to adapt the sampling process of expert networks. As observed in Fedus et al. (2021), directly sampling $D$ from $\pi$ leads to notable performance degradation (also discussed in Section 5.3). In our study, we observe an important property of the sampling process used in the Switch Transformer setting and suggest it to be the major issue with the original softmax sampling.

Marking $\boldsymbol{\theta}^* := \max_{I_i} \boldsymbol{\theta}_{I_i}$, in Switch Transformer, $I_i$ will never be sampled if:

$$\boldsymbol{\theta}^* - \boldsymbol{\theta}_{I_i} > r \cdot (|\boldsymbol{\theta}^*| + |\boldsymbol{\theta}_{I_i}|).$$

In other words, the distribution of $D$ in switch Transformer is masked: small probabilities would directly drop to zero once the corresponding logits hit a threshold. In our experiments, we observe that such sparse distribution plays a crucial role in the success of MoE and conduct more empirical discussions in the experiment section (Section 5.3).

**Adapting Expert Sampling for MoE Training.** Guided by our analyses, we deploy a sampling process that is differentiable as sampling from $\pi$, while sharing some important properties with Switch Transformer. Specifically, we changed the computation of $\pi$ from $\pi_i = \text{softmax}(\boldsymbol{\theta})_i = \frac{\exp(\boldsymbol{\theta}_i)}{\sum_{j=1}^n \exp(\boldsymbol{\theta}_j)}$ to

$$\pi_i = \frac{\exp(\boldsymbol{\theta}_i) \cdot \Delta_i}{\sum_{j=1}^n \exp(\boldsymbol{\theta}_j) \cdot \Delta_j}, \text{ where } \Delta_j = \delta(\boldsymbol{\theta}^* - \boldsymbol{\theta}_{I_i} \leq r \cdot (|\boldsymbol{\theta}^*| + |\boldsymbol{\theta}_{I_i}|)). \tag{9}$$

In other words, we apply a mask to the softmax function, in order to sample only from experts that are not masked by the Switch Transformer. This adaptation allows MoE to be trained with both sparse expert sampling and sound gradient approxiamtion, we observe it leads to a significant performance boost.

**Empirical Benefits on Expert Sampling Adaptation.** As elaborated in Section 5.3, we conduct comparisons with Switch and SparseMixer-ablation-2. Both are based on the first-order approximation as discussed in Section 3.3. The difference between these two are:

- SparseMixer-ablation-2 conducts expert sampling as in Equation 9 and uses a first-order approximation of $\nabla_0$ for parameter updates.
- Switch conducts expert sampling as in Equation 8, downscales the routing gradient by 0.5 (as in Section 3.3), thus adding additional bias to the first-order approximation.

As in Figure 1, the SparseMixer-ablation-2 method achieves consistent performance gain to Switch Transformer. applies the abovementioned sampling process to Switch Transformer, and.(see Section 5.3 for more details).

## 4 TOWARDS SECOND-ORDER ACCURACY WITH NEGLIGIBLE OVERHEADS

The literature on numerical methods for differential equations shows that it is possible to achieve higher-order accuracy *without computing higher-order derivatives*. Correspondingly, we aim to provide better gradient approximation with negligible computation overheads.

### 4.1 Achieving Second-Order Accuracy with the Mid-point Method

To furnish accurate gradient approximations, we employ a second-order ODE method, the mid-point method (briefly introduced in Appendix A). Specifically, $\widehat{\nabla}_{\text{SparseMixer-2rd}}$ is a second-order approximation of $\nabla$, where

$$\widehat{\nabla}_{\text{SparseMixer-2rd}} := 2 \cdot \frac{\partial g(\frac{\pi_D f_D(\boldsymbol{x})}{2})}{\partial W_r}.$$

To demonstrate the connection between $\widehat{\nabla}_{\text{SparseMixer-2rd}}$ and the mid-point method, we employ the mid-point method to approximate $g(\pi_{\boldsymbol{I}_i} f_{\boldsymbol{I}_i}) - g(\boldsymbol{0})$ as $g'(\frac{\pi_{\boldsymbol{I}_i} f_{\boldsymbol{I}_i}}{2}) \cdot \pi_{\boldsymbol{I}_i} f_{\boldsymbol{I}_i}$, which also requires only the output of one expert. Similarly, it is easy to note:

$$\nabla_0 \overset{\text{mid-point}}{\approx} \sum_{\boldsymbol{I}_i} g'(\frac{\pi_{\boldsymbol{I}_i} f_{\boldsymbol{I}_i}}{2}) \cdot \pi_{\boldsymbol{I}_i} f_{\boldsymbol{I}_i} \cdot \frac{\partial \pi_{\boldsymbol{I}_i}}{\partial W_r} = E_{\boldsymbol{D} \sim \boldsymbol{\pi}}[2 \cdot \frac{\partial g(\frac{\pi_D f_D(\boldsymbol{x})}{2})}{\partial W_r}] = E[\widehat{\nabla}_{\text{SparseMixer-2rd}}].$$

Notably, it is feasible to employ more advanced ODE solvers like RKF4 and approximate $\nabla_0$ with even higher-order accuracy (Fehlberg, 1969). In our experiments, we observe that the mid-point method is accurate enough and decide to stick to the mid-point method for simplicity.

### 4.2 Balancing Router Training and Expert Training

**Trade-off Behind Applying the Mid-Point Method.** Comparing to $\widehat{\nabla}_{\text{SparseMixer -1st}}$, $\widehat{\nabla}_{\text{SparseMixer-2rd}}$ provides better gradient estimation for router training. However, $\widehat{\nabla}_{\text{SparseMixer-2rd}}$ causes additional difficulties for expert training.

Specifically, $\widehat{\nabla}_{\text{SparseMixer-2rd}}$ requires to change the MoE output from $\boldsymbol{y} \leftarrow \pi_D f_D(\boldsymbol{x})$ to $\boldsymbol{y} \leftarrow \frac{\pi_D f_D(\boldsymbol{x})}{2}$. Intuitively, this change leads to two gaps:

1. A gap between the training ($\boldsymbol{y} \leftarrow \frac{\pi_D f_D(\boldsymbol{x})}{2}$) and the inference ($\boldsymbol{y} \leftarrow \pi_D f_D(\boldsymbol{x})$).

2. A gap between estimating $\nabla_0$ ($\boldsymbol{y} \leftarrow \frac{\pi_D f_D(\boldsymbol{x})}{2}$) and $\nabla_1$ ($\boldsymbol{y} \leftarrow \pi_D f_D(\boldsymbol{x})$).

In other words, applying mid-point method would lead to a better approximation of $\nabla_0$, at the cost of additional bias in computing $\nabla_1$. Similarly, as discussed in Section 5.4, such gap creates significant obstacles for MoE training.

**Hybrid Gradient Estimation: SparseMixer.** We notice that $\boldsymbol{D}$ is assigned as $\boldsymbol{D} \leftarrow \arg\max_{\boldsymbol{I}_i} \boldsymbol{\pi}_{\boldsymbol{I}_i}$ during the inference, instead of being sampled from $\boldsymbol{\pi}$. Thus, it would be sufficient to close the gap by only applying $\widehat{\nabla}_{\text{SparseMixer-2rd}}$ when $\boldsymbol{D} \neq \arg\max_{\boldsymbol{I}_i} \boldsymbol{\pi}_{\boldsymbol{I}_i}$. Accordingly, we propose SparseMixer to balance router training and expert training:

$$\widehat{\nabla}_{\text{SparseMixer}} := (1 - \delta_{\boldsymbol{D}})\widehat{\nabla}_{\text{SparseMixer-2rd}} + \delta_{\boldsymbol{D}}\widehat{\nabla}_{\text{SparseMixer-1st}}, \text{ where } \delta_{\boldsymbol{D}} = \begin{cases} 1, & \text{if } \boldsymbol{D} = \arg\max_{\boldsymbol{I}_i} \boldsymbol{\pi}_{\boldsymbol{I}_i} \\ 0, & \text{otherwise} \end{cases}.$$

**Additional Sampling Adaptation.** Since the value of $\boldsymbol{\pi}$ will be different after applying the mask (which impacts the gradient magnitude of other components), we further changed the output of the MoE layer from $\boldsymbol{\pi}_{\boldsymbol{D}} \cdot f_{\boldsymbol{D}}(\boldsymbol{x})$ to $\boldsymbol{\omega} \cdot \boldsymbol{\pi}_{\boldsymbol{D}} \cdot f_{\boldsymbol{D}}(\boldsymbol{x})$, where $\boldsymbol{\omega}$ is trainable and is initialized as the $\boldsymbol{1}$ vector. Intuitively, $\boldsymbol{\omega}$ can be viewed as an adaptation on the learning rate for training expert networks. Note that, $\boldsymbol{\omega}$ can be re-parameterized into the feedforward layer after training.

**Computational Efficiency of SparseMixer .** $\widehat{\nabla}_{\text{SparseMixer}}$ does not require Hessian or other second-order derivatives, thus having negligible computation overheads. Also, it is worth mentioning that $\widehat{\nabla}_{\text{SparseMixer}}$ has the same order of computation complexity and memory complexity with only estimating $\nabla_1$ (i.e., the current practice of MoE training neglects $\nabla_0$ directly). Empirical verification is discussed in Section 5.5, which matches our analyses here.

At the same time, similar to $\widehat{\nabla}_{\text{ST}}$, our proposed algorithm can be easily integrated with popular library like PyTorch, making it easy to be integrated with existing algorithms.

## 5 EXPERIMENTS

Here, we conduct experiments on both pretraining and neural machine translation tasks.

### 5.1 EXPERIMENT SETTING

We closely follow the experiment setting of the existing study. Due to the constraint of computation resources, we left MoE related hyper-parameters untuned in all settings, i.e., jitter ($r$) is set to 0.1 and load balance loss ratio is set to 0.01 (Fedus et al., 2021). Detailed experiment configurations are elaborated in Appendix B.

### 5.2 APPLYING SPARSEMIXER ON SWITCH TRANSFORMER

**NMT on WMT'14 En-De.** We visualized the training curve in Figure 1 and summarized the BLEU score in Table 1. Regarding both convergence speed and the final performance, Switch+SparseMixer consistently outperforms Switch in all five settings. Notably, Switch+SparseMixer matches the training performance of Switch with about *50% less training updates when $N \in \{4, 6, 8\}$ and about *40% less training updates when $N \in \{2, 16\}$*.

We can observe that, with more experts, MoE models achieve lower training loss with a worse BLEU score. Specifically, although Switch Transformer achieves better training performance, its final performance (BLEU score) never outperforms the Dense model, regardless of how many experts it has. We believe it requires more data to fully unleash the potential of MoE and suggest this phenomenon indicates that MoE models are prone to overfitting (Zuo et al., 2022).

Meanwhile, without changing hyper-parameters or model architectures, the downstream performance of Switch + SparseMixer outperforms both Dense and Switch, when $N \in \{2, 4\}$. Specifically, SparseMixer improves the performance of Switch from 28.17 to 28.72 (when $N = 2$) and from 28.05 to 28.61 (when $N = 4$). This phenomenon implies that, with the help of SparseMixer, a sound gradient estimator, MoE learns an expert routing that generalizes better.

**Pretraining.** Following previous work (Dong et al., 2023), we visualized the training curve in Figure 6 and summarized the fine-tuning results in Table 2. Regarding both convergence speed and downstream performance, Switch+SparseMixer consistently outperforms Switch in all settings. Also, similar to the experiments on machine translation, we observe that MoE models are easier to overfit and both settings achieve the best downstream performance with two experts.

Also, it is worth mentioning that, while Switch Transformer only outperforms the dense model when the number of experts is set to 2, Switch + SparseMixer consistently outperforms the Dense model in all four settings. This phenomenon further verifies our intuition that SparseMixer facilitates MoE models with better expert router training, thus having the resulting model to generalize better.

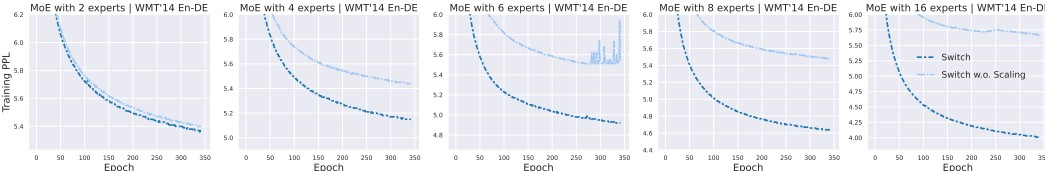

Figure 2: Comparison between *Switch Transformer* and *Switch Transformer without Scaling*.

Table 1: BLEU score on WMT'14 En-De ($N$ refers to the number of experts).

| | Dense | Mixture-of-Expert | | | | |
|---|---|---|---|---|---|---|
| | | $N = 2$ | $N = 4$ | $N = 6$ | $N = 8$ | $N = 16$ |
| Transformer-base | 28.33 | / | / | / | / | / |
| Switch | / | 28.17 | 28.05 | 27.96 | 27.99 | 27.81 |
| Switch+SparseMixer | / | **28.72** | **28.61** | **28.32** | **28.12** | **28.08** |

Table 2: Results on the GLUE development set. S refers to Switch and S+S refers to Switch+SparseMixer. AVG is the average score across eight tasks.

| $N$ | Model | AVG | MNLI-(m/mm) (Acc.) | QQP (Acc.) | QNLI (Acc.) | SST-2 (Acc.) | CoLA (Mat. Corr.) | RTE (Acc.) | MRPC (Acc.) | STS-B (Spear. Corr.) |
|---|---|---|---|---|---|---|---|---|---|---|
| 1 | Dense | 87.37 | 88.72/88.40 | 91.90 | 93.36 | 93.35 | 68.71 | 82.31 | 89.95 | 90.83 |
| 2 | S | 87.62 | 88.55/88.34 | 91.86 | 93.52 | 94.27 | 67.90 | 83.76 | 90.69 | 90.52 |
|   | S+S | 88.31 | 89.06/88.78 | 91.98 | 93.54 | 94.38 | 69.96 | 85.20 | 91.67 | 90.81 |
| 4 | S | 87.02 | 88.12/88.40 | 91.73 | 93.21 | 93.92 | 70.89 | 77.26 | 90.44 | 90.49 |
|   | S+S | 87.63 | 88.97/88.41 | 91.92 | 93.54 | 94.04 | 71.00 | 80.87 | 90.69 | 90.72 |
| 8 | S | 87.27 | 88.43/88.22 | 91.78 | 93.23 | 94.84 | 68.06 | 80.87 | 90.44 | 90.62 |
|   | S+S | 87.71 | 88.69/88.47 | 92.03 | 93.41 | 94.15 | 69.00 | 83.76 | 89.95 | 90.81 |

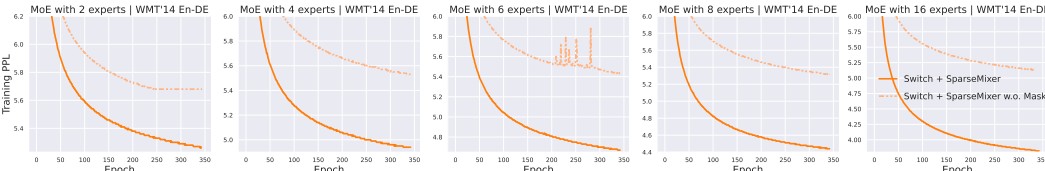

Figure 3: Comparison between *SparseMixer* and *SparseMixer without applying mask to sampling*.

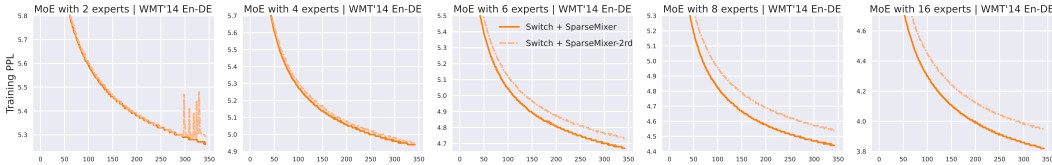

Figure 4: Comparison between *SparseMixer* and *SparseMixer-2rd*.

## 5.3 DISCUSSIONS

Here, we conduct experiments to discuss our modeling of the MoE layer as in Section 2.

**Importance of Scaling Expert Outputs with Gating Networks.** One important design detail of MoE is to scale the output of the expert network with the gating network. Specifically, the output of the MoE layer is computed as $y \leftarrow \pi_D f_D(x)$, instead of $y \leftarrow f_D(x)$. This scaling design greatly facilitates the derivation of SparseMixer in Section 3, and inspires the introduction of $\omega$ (further discussed in Section 5.4). Here, we empirically demonstrate that this scaling design also plays an important role in Switch Transformer.

Specifically, we conduct experiments with a variant of Switch Transformer, i.e., Switch w.o. Scaling, which sets the output of the MoE layer as $y \leftarrow f_D(x)$. We apply this Switch variant on WMT'14 En-De and visualize the training curve in Figure 2. Switch ($y \leftarrow \pi_D f_D(x)$) significantly outperforms this variant ($y \leftarrow f_D(x)$). Also, we can observe that, when the number of experts is set to 6, using this variant would lead to additional training instability, which further demonstrates the importance of the scaling design.

**Importance of Applying Mask to Softmax.** In Section 3.3, we identify that the sampling in Switch Transformer plays an important role in the success of Switch Transformer. As discussed in Fedus et al. (2021), directly using softmax sampling would lead to an inferior performance.

Here, we demonstrate that this masked softmax sampling also plays an important role in Switch + SparseMixer. Specifically, we conduct experiments with a variant of SparseMixer, i.e., SparseMixer w.o. Mask, which computes $\pi_i \leftarrow \text{softmax}(\theta)_i$. We apply SparseMixer w.o. Mask on WMT'14 En-De and visualize the training curve in Figure 3. SparseMixer ($\pi_i \leftarrow \frac{\exp(\theta_i)\cdot\Delta_i}{\sum_{j=1}^{n}\exp(\theta_j)\cdot\Delta_j}$) significantly outperforms this variant ($y \leftarrow \text{softmax}(\theta)_i$). Also, we can observe that, when the number of experts is set to 6, using this variant would lead to additional training instability, which further demonstrates the importance of applying mask to softmax.

Table 3: Average Training Time Cost (s/update). $N$ refers to the number of experts.

| | WMT'14 En-De | | | | | LM Pre-training | | |
| --- | --- | --- | --- | --- | --- | --- | --- | --- |
| | $N = 2$ | $N = 4$ | $N = 6$ | $N = 8$ | $N = 16$ | $N = 2$ | $N = 4$ | $N = 8$ |
| Switch | 0.32 | 0.33 | 0.34 | 0.36 | 0.40 | 1.87 | 1.90 | 1.98 |
| Switch + SparseMixer | 0.32 | 0.33 | 0.34 | 0.36 | 0.40 | 1.87 | 1.90 | 1.98 |

## 5.4 ABLATION

**Importance of Balancing Expert Learning and Routing Learning.** While SparseMixer-2rd provides better gradient approximation for expert routing, it creates a gap between training and inference. To demonstrate the importance of balancing router training and expert training, we conduct experiments on applying SparseMixer-2rd on WMT'14 En-De. As visualized in Figure 4, SparseMixer consistently outperforms SparseMixer-2rd in all cases. Also, SparseMixer-2rd exhibits training instability when setting the number of experts to 2.

**Mid-point Method and $\omega$ Scaling.** To better understand the benefit introducing $\omega$ (as in Section 3.3) and make comparisons with SparseMixer-1st, we conduct additional ablation studies on WMT'14 En-De. Specifically, we consider two SparseMixer variants:

- ablation-1 removes $\omega$ from SparseMixer (i.e., changes the output of Switch + SparseMixer from $\omega \cdot \pi_D \cdot f_D(x)$ to $\pi_D \cdot f_D(x)$).
- ablation-2 further replaces the mid-point method with the forward Euler method in SparseMixer-ablation-1, i.e., $\widehat{\nabla}_{\text{SparseMixer-1st}}$ is employed as the gradient estimator and $\omega$ is removed.

We apply these two variants to WMT'14 En-De. As in Figure 1, both variants outperform the baseline. The results further verified our intuition that $\omega$ facilitates MoE training by alleviating the impact of applying masks. Also, it shows that integrating the mid-point method helps to better approximate expert routing gradient.

## 5.5 EFFICIENCY

We summarized the average time cost per update in Table 3. Switch+SparseMixer achieves an identical average time cost with Switch in all eight settings. This shows that the computation overheads of SparseMixer are negligible.

To better understand the computation overhead brought by SparseMixer, we compute the floating point operations (FLOPS) for one forward propagation and one backward propagation. We visualized the FLOPS ratio of Switch and Switch+SparseMixer of various number of experts in Figure 5. It shows the computation overhead brought by SparseMixer only composes up to 0.1% of the total training FLOPS, for MoE models with up to 16384 experts.

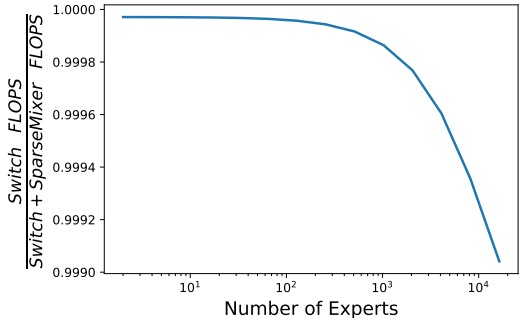

Figure 5: The ratio of Switch Training FLOPS and Switch+SparseMixer Training FLOPS. The FLOPS are computed for ELECTRA-base training with one 512-token sequence.

## 6 CONCLUSION

In this study, we present SparseMixer to move beyond ST and bridge the gap between sparse MoE routing and backpropagation. Rooted in a numerical ODE framework, SparseMixer harnesses the mid-point method, a second-order ODE solver, to deliver precise gradient approximations with negligible computational overhead. In our experiments on both neural machine translation and pre-training tasks, SparseMixer not only accelerates training convergence by up to two times but also facilitates MoE with properly trained expert routing. Remarkably, while Switch Transformer underperforms the dense model in all three pretraining settings, incorporating SparseMixer as the gradient estimator allows the resulting MoE models to consistently outperform the dense model.

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

## A  FORWARD EULER METHOD AND HEUN'S METHOD

For simplicity, we consider a simple function $g(x) : \mathcal{R} \to \mathcal{R}$ that is three times differentiable on $[t_0, t_1]$. Now, we proceed to a simple introduction to approximate $\int_{t_0}^{t_1} g'(x)dx$ with the Forward Euler Method and the Heun's Method. For a detailed introduction to numerical ODE methods, please refer to Ascher & Petzold (1998).

**Forward Euler Method.**  Here, we approximate $g(t_1)$ with the first-order Taylor expansion, i.e., $g(t_1) = g(t_0) + g'(t_0) \cdot (t_1 - t_0) + O((t_1 - t_0)^2)$, then we have $\int_{t_0}^{t_1} g'(x)dx \approx g'(t_0)(t_1 - t_0)$. Since we used the first-order Taylor expansion, this approximation has first-order accuracy.

**Heun's Method.**  First, we approximate $g(t_1)$ with the second-order Taylor expansion:

$$g(t_1) = g(t_0) + g'(t_0) \cdot (t_1 - t_0) + \frac{g''(t_0)}{2} \cdot (t_1 - t_0)^2 + O((t_1 - t_0)^3). \qquad (10)$$

Then, we show that we can match this approximation by combining the first-order derivatives of two samples. Taylor expanding $g'(\frac{t_1 + t_0}{2})$ to the first-order, we have:

$$g'(\frac{t_1 + t_0}{2}) = g'(t_0) + g''(t_0) \cdot \frac{t_1 - t_0}{2} + O((t_1 - t_0)^2)$$

Therefore, we have:

$$g(t_0) + g'(\frac{t_1 + t_0}{2})(t_1 - t_0) = g(t_0) + g'(t_0) \cdot (t_1 - t_0) + \frac{g''(t_0)}{2} \cdot (t_1 - t_0)^2 + O((t_1 - t_0)^3).$$

It is easy to notice that the right-hand side of the above equation matches the second-order Taylor expansion of $g(t_1)$ as in Equation 10. Therefore, the above approximation (i.e., approximating $g(t_1) - g(t_0)$ as $g'(\frac{t_1+t_0}{2})(t_1 - t_0)$) has second-order accuracy.

**Connection to $f(\boldsymbol{I}_i) - f(\boldsymbol{0})$.** By setting $g(x) = f(x \cdot \boldsymbol{I}_i)$, we have $g(1) - g(0) = f(\boldsymbol{I}_i) - f(\boldsymbol{0})$. Then, it is easy to notice that the forward Euler Method approximates $f(\boldsymbol{I}_i) - f(\boldsymbol{0})$ as $\frac{\partial f(\boldsymbol{I}_i)}{\partial \boldsymbol{I}_i} \boldsymbol{I}_i$ and has first-order accuracy. Also, the mid-point method approximates $f(\boldsymbol{I}_i) - f(\boldsymbol{0})$ as $\frac{\partial f(\boldsymbol{I}_i/2)}{\partial \boldsymbol{I}_i/2} \boldsymbol{I}_i$ and has second-order accuracy.

Table 4: GLUE task descriptions and statistics. The second and fourth column denotes the number of training examples and the number of classes. Note that STS-B is a regression task.

| Corpus | |Train| | |Label| | Task | Metric(s) | Domain |
|---|---|---|---|---|---|
| | | | Single-Sentence Classification | | |
| CoLA | 8.5k | 2 | acceptibility | Matthews corr. | misc. |
| SST-2 | 67k | 2 | sentiment | accuracy | movie reviews |
| | | | Sentence Similarity/Paraphrase | | |
| MRPC | 3.7k | 2 | paraphrase | accuracy | news |
| STS-B | 5.7k | - | similarity | Spearman corr. | misc. |
| QQP | 364k | 2 | similarity | accuracy | social QA questions |
| | | | Natural Language Inference (NLI) | | |
| MNLI | 393k | 3 | NLI | (mis)matched acc. | misc. |
| QNLI | 108k | 2 | QA/NLI | accuracy | Wikipedia |
| RTE | 2.5k | 2 | NLI | accuracy | misc. |
| WNLI | 634 | 2 | coreference/NLI | accuracy | fiction books |

Table 5: Hyperparameter search space in fine-tuning.

| Hyperparameters | Base |
|---|---|
| Sequence Length | 256 |
| Optimizer | Adam |
| Peak Learning Rate | {5e-5,1e-4, 3e-4} |
| Max Epochs | {2,3,5,10} |
| Batch size | {16, 32} |
| Learning rate decay | Linear |
| Weight Decay | {0, 0.01} |
| Warm-up Proportion | {6 %, 10 %} |
| Adam $\epsilon$ | 1e-6 |
| Adam $(\beta_1, \beta_2)$ | $(0.9, 0.98)$ |
| Gradient Clipping | 1.0 |
| Dropout | 0.1 |

## B EXPERIMENT SETTING

### B.1 NEURAL MACHINE TRANSLATION

**Problem Setting.** Our experiments are based on the fairseq package (Ott et al., 2019). As to pre-processing, we follow the public released script from previous work (Lu et al., 2020), and conduct evaluations on the provided 'newstest14' file. More details can be found in Bojar et al. (2014).

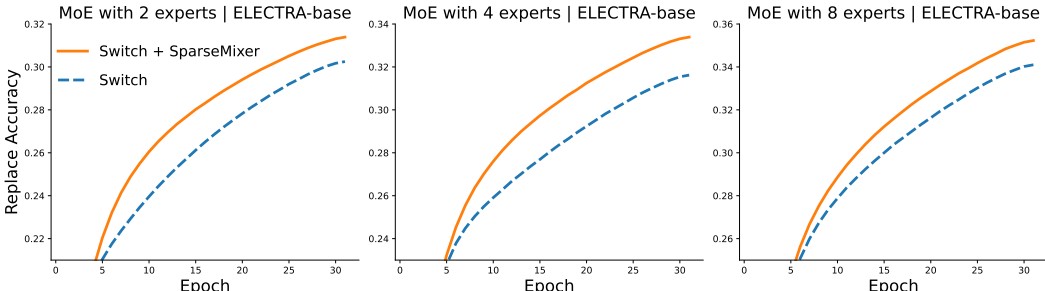

Figure 6: Training curves of Switch Transformer on ELECTRA-base training.

**Model Architecture.** As to model specifics, we directly adopt the Transformer-base model on the WMT'14 En-De datasets. Specifically, we use encoder-decoder Transformer models with 6 encoder layers, 6 decoder layers, 512-dimension word embedding, 8-head attentions, and 2048-dimension feed-forward layers. Following Fedus et al. (2021), we apply MoE layers at every other feed-forward layers, set jitter to 0.1, and configure load balance ratio as $1 \cdot 10^{-2}$. As the number of experts, we consider 5 different settings, i.e., $N \in \{2, 4, 6, 8, 16\}$. Label smoothed cross-entropy is used as the objective function with the uncertainty set as 0.1 (Szegedy et al., 2016).

**Training Settings.** We mostly followed (Liu et al., 2020a) for training settings. Specifically, we use Adam as the optimizer set $(\beta_1, \beta_2)$ as $(0.9, 0.98)$, use inverse sqrt learning rate scheduler with a warmup phrase (8000 steps). All dropout ratios (including activation dropout and attention dropout) are set to 0.1. The maximum learning rate is set to $7 \cdot 10^{-4}$ and the maximum token number per batch is set to $2^{17}$. We conduct training for $4 \cdot 10^5$ updates and report the performance of the last checkpoint and the checkpoint with the lowest development loss.

## B.2 PRE-TRAINING

**Pre-training Setup.** We follow the standard settings for training Base models (Clark et al., 2020; Bajaj et al., 2022; Dong et al., 2023), Specifically, we employ Wikipedia and BookCorpus (Zhu et al., 2015) for pre-training and set the sequence length to 512, which leads to 16 GB of texts and 256M samples. We use a cased sentence piece BPE vocabulary of 128K tokens following He et al. (2020), and conduct pre-training for 125K updates with a batch size of 2048 sentences.

**Model Architecture.** Our main model (discriminator) setting follows the BERT$_{base}$ architecture (Devlin et al., 2019). Specifically, the model has 12 layers, 768-dimension embedding, and 12-head attention. As to the feed-forward networks, we set the number of hidden state dimensions to 3076. Following Bajaj et al. (2022) and Dong et al. (2023), we further enhanced the model with the T5 relative position encoding (Raffel et al., 2019) and use 32 bins. We set dropout as 0.1 and employ Admin (Liu et al., 2020b) for model initialization to stabilize the training. Following Fedus et al. (2021), we apply MoE layers at every other feed-forward layers, set jitter to 0.1, and configure load balance ratio as $1 \cdot 10^{-2}$. As the number of experts, we consider 3 different settings, i.e., $N \in \{2, 4, 8\}$. As to the auxiliary model, we follow previous works (Clark et al., 2020; Bajaj et al., 2022) to set the size of the auxiliary model (generator) to be 4 layers.

**Optimization.** We configure the optimizer as Adam, $(\beta_1, \beta_2)$ as $(0.9, 0.98)$, weight decay as 0.01, the loss weight as 50, the peak learning rate as $5e - 4$, and the warmup steps as 10K.

**Downstream evaluation setup.** We conduct evaluation on downstream tasks following the setup in previous works (Bajaj et al., 2022). Specifically, we conduct single-task, single-model fine-tuning on the GLUE (Wang et al., 2018) benchmark. As summarized in the Appendix (Table 4), GLUE includes 9 subtasks. Following Liu et al. (2019), we conduct a grid-search on hyper-parameters and report the best performance for both Switch and Swith + SparseMixer. The complete search space is included in Appendix (Table 5).

