# OpenReview forum: "Sparse Backpropagation for MoE Training"
_ICLR.cc/2024/Conference — Submitted to ICLR 2024_

### Official Review · Reviewer_yUXn · 2023-10-30

**Soundness:** 3 good
**Presentation:** 3 good
**Contribution:** 2 fair
**Rating:** 3
**Confidence:** 3

**Summary:**

This work investigates the problem non-differentiable of the expert selection, which is relevant in today Sparse Mixture of Experts (SMoE) implementations. Motivated from ODE methods, the proposed method, SparseMixer, derives a mid-point method to approximate the gradient with neglectable overhead. Experiments on pre-training, finetuning, and machine translation show that SparseMixer consistently outperforms SwitchTransformer and Dense training.

**Strengths:**

- SMoE has shown to be a promising direction to massively scale transformer models. This works investigates the non-differentiable problem of selection experts, which is at heart of existing SMoE strategies.
- The proposed method is sound.
- The empirical results are encouraging where SparseMixer consistently outperforms SwitchTransformer with neglectable overhead.

**Weaknesses:**

## Major concern - limited broad impact
Although the method is sound, it seems to be only applicable to the case of choosing only $1$ expert per layer. However, most modern architecture employ a TopK function that choose $k=2$ experts, which has shown to achieve better results, e.g.[A]. Thus, despite the encouraging results, it may not bring immediate impact.

## Major concern - contribution of $\omega$-scaling
$\omega$-scaling essentially makes the network deeper, thus using it improves the performance is easy to understand. However, it also makes comparing with SwitchTransformer to be unfair because of the additional trainable parameters. How would SwitchTransformer+$\omega$-scaling performs against SparseMixer?

## Major concern - complexity analysis
I found the results in Table 3 to be questionable where SwitchTranformer and SparseMixer have identically average training costs. However, SparseMixer introduces several components such as $\omega$-scaling, computing the gradient $\nabla_0$, all of which contribute to the forward and backward computation. Thus, I think reporting the **total** training is more accurate to understand the overhead of SparseMixer.

## Minor concern - limited baselines
The number of baselines considered is quite limited. The authors also considered a simple naive Transformer architecture. Other advanced architecture such as GLAM [A] and other SMoE strategies such as XMoE [B] should be included to make the experiment more comprehensive.

## Minor concern - presentation
- I suggest the author to replace "MoE" to "SMoE" as MoE usually refers to using all experts. This work consider the selecting only 1 expert, thus adding the "sparse" keyword would make the presentation clearer.
- Best results are not highlighted in Table 2.

[A] Du, Nan, et al. "Glam: Efficient scaling of language models with mixture-of-experts." International Conference on Machine Learning. PMLR, 2022.

[B] Chi, Zewen, et al. "On the representation collapse of sparse mixture of experts." Advances in Neural Information Processing Systems 35 (2022): 34600-34613.

**Questions:**

- Can SparseMixer be extended to $k>1$ easily?
- Please clarify the empirical contribution of $\omega$-scaling, i.e. comparing SparseMixer without $\omega$-scaling with SwitchTransformer **OR** SparseMixer with SwitchTransformer + $\omega$-scaling.
- Please clarify the overhead in SparseMixer in Table 3 and report the total training time.

---

> ### Author Response · Authors · 2023-11-23
> **Rebuttal by Authors**
>
> Thank you for your comprehensive feedback. We appreciate the opportunity to clarify and expand on the strengths of our approach, with further elaborations to be included in the final paper.
>
> **Reply to presentation:**
>
> We appreciate your suggestion regarding the term. We have retained "MoE" in our manuscript, as it is a widely recognized term in the literature, predominantly referring to sparsely activated networks [1,2,3]. Also, we would like to acknowledge that "SMoE" could potentially lead to confusion, as it might be misconstrued as Soft MoE [4].
> 1. Adaptive mixtures of local experts. Neural Computation 1991
> 2. Hierarchical mixtures of experts and the em algorithm. Neural Computation 1994
> 3. Switch Transformers: Scaling to Trillion Parameter Models with Simple and Efficient Sparsity. 2021
> 4. From Sparse to Soft Mixtures of Experts. NeurIPS 2022
>
> **Reply to Efficiency Evaluation:**
>
> Table 3 in our initial submission summarized the wall-clock time per epoch (excluding checkpoint saving) and took into account all components of SparseMixer, including omega scaling. To provide further clarity, we have now added Figure 5, which compares the training FLOPs ratio between Switch and Switch+SparseMixer configurations.
>
> **Reply to Omega Scaling:**
> To evaluate the impact of Omega scaling, we conducted a specific ablation study, which is presented as ablation-1 SparseMixer variant in Figure 1 of our submission. This variant excludes Omega scaling from SparseMixer, allowing for a direct comparison of its effects.
>
> **Reply to Research Scope:**
> Early on, neural networks were trained using a variety of update rules, such as the classical Winnow algorithm and the Perceptron algorithm. Given that these algorithms were largely handcrafted, generalizing these update rules to accommodate deeper and more intricate neural networks posed significant challenges.
>
> The introduction of the backpropagation algorithm and the stochastic gradient descent algorithm re-anchored the foundation of neural networks to the principles of gradient descent and the chain rule, mirroring the foundational ideas behind least squares computation. This paradigm shift has been instrumental, with the vast majority of neural network research post-2000 relying on this update rule.
>
> However, backpropagation cannot be directly applied to sparse model training. Accordingly, many attempts have been made to derive ad-hoc proxy for gradient computation and gradient descent. However, there is no guarantee the resulting estimation can be viewed as a gradient approximation, and it is not clear doing gradient descent with these proxies can lead to a descent.
>
> Our study reveals the underlying connection between the current MoE training practice and gradient approximation. It not only re-anchored MoE training as gradient descent at-scale paradigm, but also provides guidance on how to obtain better gradient estimation.
>
> Since our algorithm provides better gradient estimation, the gradient-based optimizer would be more effective at more training, thus leading to better model training (accelerating model training by up to 2 times).
>
> **Reply to Experiment Design:**
>
> In our submission, we adhered to the experiment design of the existing study, focusing on normal-scale problems for controllability and resource efficiency. For example, choosing ELECTRA-base as the pre-training task allows us to conduct experiments with multiple settings. As a comparison, typical ELECTRA-base model training takes 125k steps and typical BERT-base model training takes 1M steps.
>
> While our method has potential applications in different scales and MoE gating methods, the focus of this study is on foundational gradient estimation. Scaling and adapting our method for varied MoE paradigms presents both research and engineering challenges. We aim to explore these in future work, considering our current computational resource constraints.
>
> **Reply to SparseMixer Application on Top-k Routing:**
>
> SparseMixer is for estimating gradients involving discrete random variables with only sparse backpropagation. In principle, it is applicable to other MoE routing algorithms. The only bottleneck for applying SparseMixer is that it requires the expert index is explicitly sampled from a parameterized distribution, which requires us to apply some adaptation to Switch Transformer / top-1 routing.

---

> ### Comment · Reviewer_yUXn · 2023-12-04
> **Acknowledgement**
>
> I appreciate the authors' efforts addressing my concern. However, after going through the revision and the posted comments, I am not entirely convinced that this work is ready for publications. My concerns regarding the contributions and presentation (minor) still remains. Particularly:
> - The newly added results in Figure 1 are poorly discussed, the Figure is placed 3 pages before it is mentioned.
> - The contribution of $\omega-$scaling is poorly discussed. Given Figure 1, I could argue that in many cases, most of the contribution comes from the $\omega-$scaling while the proposed method only contribute marginally.
> - The authors failed to report the total training time or explain why SparseMixer could achieve the same complexity despite introducing several computational steps.
> - The replies to regarding the extension to $k\ge 2$ or application on different architectures are not satisfactory. I expected the authors to outline some steps or challenges to extend to cases $k\ge 2$ and provide a minimal experiment/insight with a different architecture. However, none of such results are presented.
> - Naming convention (minor): the name Sparse Mixer has been used in an existing work [A] for a different method.
>
> Although the approach is somewhat interesting, I believe the current draft still has many issues. Thus, I keep my original rating of this work and don't endorse the paper to be accepted at its current state.
>
> [A] Lee-Thorp, James, and Joshua Ainslie. "Sparse Mixers: Combining MoE and Mixing to build a more efficient BERT." arXiv preprint arXiv:2205.12399 (2022).

---

### Official Review · Reviewer_tyBo · 2023-10-31

**Soundness:** 3 good
**Presentation:** 2 fair
**Contribution:** 2 fair
**Rating:** 6
**Confidence:** 3

**Summary:**

This paper proposes SparseMixer to move beyond discrete and bridge the gap between sparse MoE routing and backpropagation. SparseMixer utilizes a second-order ODE solver to deliver precise gradient approximations. Their experiments show SparseMixer not only accelerates training convergence by up to two times but also facilitates MoE with properly trained expert routing.

**Strengths:**

1. The pre-training of LLMs is costly, and MoE is one promising sparse training method to reduce the training overhead. The paper makes some contributions to improving the backpropagation of MoE Training.

2. The experiment results show the effectiveness of the proposed methods on a specific model.

**Weaknesses:**

My concern includes two aspects:

1. The experiment is kind of weak since there are also some other MoE architecture and other base models, while the authors only focus on Switch. Currently, the author only focuses on a simplified setting of the switch Transformer layer (Fedus et al., 2021). However, there are also other popular MoE architectures, e.g. (Shazeer et al., 2017; Lepikhin et al., 2020; Lewis et al., 2021) as mentioned in the paper, and (Yanqi et al., 2022; Nan et al., 2022) in the following.

Zhou, Yanqi, Tao Lei, Hanxiao Liu, Nan Du, Yanping Huang, Vincent Zhao, Andrew M. Dai, Quoc V. Le, and James Laudon. "Mixture-of-experts with expert choice routing." Advances in Neural Information Processing Systems 35 (2022): 7103-7114. Du, Nan, Yanping Huang,

Andrew M. Dai, Simon Tong, Dmitry Lepikhin, Yuanzhong Xu, Maxim Krikun et al. "Glam: Efficient scaling of language models with mixture-of-experts." In International Conference on Machine Learning, pp. 5547-5569. PMLR, 2022.

2. The writing of Section 3 is too fragmented and can be improved a lot. I would recommend authors to enrich Sections 3.2 and 3.2, since currently, they read like an experiment report with step-by-step procedures, instead of a well-written technical paper with good motivation and intuition of the proposed techniques.

**Questions:**

1. Can the authors explicitly present the physical running time? The results in the experiment section do not give a clear comparison.

2. What is the experiment platform, e.g. torch version or GPU model?

---

> ### Author Response · Authors · 2023-11-23
> **Rebuttal by Authors**
>
> Thank you for your comprehensive feedback. We appreciate the opportunity to clarify and expand on the strengths of our approach, with further elaborations to be included in the final paper.
>
> **Reply to weakness 1 (scope of study)**
>
> Our study primarily focuses on gradient estimation in MoE training.
>
> Early on, neural networks were trained using a variety of update rules, such as the classical Winnow algorithm and the Perceptron algorithm. Given that these algorithms were largely handcrafted, generalizing these update rules to accommodate deeper and more intricate neural networks posed significant challenges.
>
> The introduction of the backpropagation algorithm and the stochastic gradient descent algorithm re-anchored the foundation of neural networks to the principles of gradient descent and the chain rule, mirroring the foundational ideas behind least squares computation. This paradigm shift has been instrumental, with the vast majority of neural network research post-2000 relying on this update rule.
>
> However, backpropagation cannot be directly applied to sparse model training. Accordingly, many attempts have been made to derive ad-hoc proxy for gradient computation and gradient descent. However, there is no guarantee the resulting estimation can be viewed as a gradient approximation, and it is not clear doing gradient descent with these proxies can lead to a descent.
>
> Our work bridges this gap by revealing the intrinsic connection between current MoE training practices and gradient approximation. This not only repositions MoE training within the gradient descent paradigm at scale but also guides more accurate gradient estimation.
>
> **Reply to weakness 1 (significance of performance gain):**
>
> We believe the improvement brought by SparseMixer is significant. Regarding training convergence, SparseMixer accelerates model training by up to two times. Regarding downstream performance, the best performance achieved by SparseMixer is 28.72 on WMT’14 EN-DE (note our architecture configuration is based on Transformer-base) and 88.31 on GLUE (note our architecture configuration is based on Electra-base).
>
> It's important to note that the improvements on GLUE are actually significant, as GLUE is a very competitive benchmark, Most leading methods on the GLUE benchmark (as in https://gluebenchmark.com/leaderboard) are based on the Electra-large setting and their improvements over the previous methods are usually within 0.2 absolute points. In our study, simply applying SparseMixer to Electra-base training without any pre-training hyper-parameter re-tuning, we achieve a consistent 0.5 absolute point improvements.
>
>
>
> While our method has potential applications in different scales and MoE gating methods, the focus of this study is on foundational gradient estimation. Scaling and adapting our method for varied MoE paradigms presents both research and engineering challenges. We aim to explore these in future work, considering our current computational resource constraints.
>
>
> **Reply to weakness 2 (Section 3 writing)**
> We appreciate your feedback on Section 3. The section has been revised for improved coherence, focusing on the rationale and intuition behind our techniques rather than merely describing experimental procedures.
>
> **Reply to question 1**
> In our submission, Table 3 summarizes the wall-clock time for the model training (measured by total training time per epoch without checkpoint saving, divided by number of updates). In our revised submission, Figure 5 summarizes the training FLOPs ratio between Switch and Switch+SparseMixer.
>
> **Reply to question 2**
> Following previous study, we used the `nvidia/pytorch:22.04-py3` docker image for the pre-training task and the `nvidia/pytorch:22.02-py3`docker image for the NMT task. The training was primarily conducted on V100 GPUs, with some fine-tuning on P40/P100 GPUs.

---

### Official Review · Reviewer_D3YK · 2023-10-31

**Soundness:** 3 good
**Presentation:** 3 good
**Contribution:** 3 good
**Rating:** 5
**Confidence:** 3

**Summary:**

The paper presents SparseMixer, a scalable gradient estimator that bridges the gap between backpropagation and sparse expert routing.  In specific, the paper presents SparseMixer to provide scalable gradient approximations for the gradients terms that are not taken into account, enabling reliable gradient estimation in MoE training. The authors demonstrate SparseMixer to Switch Transformer on both pre-training and machine translation tasks, with the claim of considerable performance gain, accelerating training convergence by up to 2 times.

**Strengths:**

1. The idea of improving gradient computation at scale to improve MoE training is novel to me.

2.The paper consistently demonstrate the impact of neglecting $\Delta_0$ in the pre-training with MoE.

3. The paper is written well and the results back up the improvement.

**Weaknesses:**

1. Straight-Through (ST) -- > straight through estimator (STE) ?

2. Please define ODE first in the abstract before using the abbr.

3. Please introduce definitions of $\Delta_0$ and $\Delta_1$.

4. It is not quite clear why the training speed improves.

5. Please demonstrate results with other MoE gating methods. as few of them tried to improve MoE training.

**Questions:**

Please see weakness.

---

> ### Author Response · Authors · 2023-11-23
> **Rebuttal by Authors**
>
> Thank you for your comprehensive feedback. We appreciate the opportunity to clarify and expand on the strengths of our approach, with further elaborations to be included in the final paper.
>
> **Reply to weakness 1 (terminology)**
> We choose ST as the abbreviation, since ST is a widely used terminology in the literature [1,2,3].
>
> [1] Bridging Discrete and Backpropagation: Straight-Through and Beyond. NeurIPS 2023
>
> [2] Training discrete deep generative models via gapped straight-through estimator. ICML 2022
>
> [3] Rao-blackwellizing the straight-through gumbel softmax gradient estimator. ICLR 2021
>
> **Reply to weakness 2 (Definition of ODE in Abstract)**
> Thanks for the suggestion! We have revised our abstract to include a clear definition of Ordinary Differential Equations (ODE) at its first mention.
>
> **Reply to weakness 3 (Definition of $\nabla_0$ and $\nabla_1$)**
> $\nabla_0$ and $\nabla_1$ are defined in Section 2 (Equation 2). In response to your feedback, we have added references in Section 3 of the revised manuscript for easier navigation and understanding.
>
> **Reply to weakness 4 (Clarification on the Mechanism of SparseMixer)**
> The improved training speed is a result of our enhanced gradient estimation.
>
> Early on, neural networks were trained using a variety of update rules, such as the classical Winnow algorithm and the Perceptron algorithm. Given that these algorithms were largely handcrafted, generalizing these update rules to accommodate deeper and more intricate neural networks posed significant challenges.
>
> The introduction of the backpropagation algorithm and the stochastic gradient descent algorithm re-anchored the foundation of neural networks to the principles of gradient descent and the chain rule, mirroring the foundational ideas behind least squares computation. This paradigm shift has been instrumental, with the vast majority of neural network research post-2000 relying on this update rule.
>
> Since backpropagation cannot be directly applied to sparse model training, many attempts have been made to derive ad-hoc proxy for gradient computation and gradient descent. However, there is no guarantee the resulting estimation can be viewed as a gradient approximation, and it is not clear doing gradient descent with these proxies can lead to a descent.
>
> Our study reveals the underlying connection between the current MoE training practice and gradient approximation. It not only re-anchored MoE training as gradient descent at-scale paradigm, but also provides guidance on how to obtain better gradient estimation.
>
> Since our algorithm provides better gradient estimation, the gradient-based optimizer would be more effective at more training, thus leading to better model training.
>
> **Reply to weakness 5 (Exploration of Other MoE Gating Methods)** While our method has potential applications in different scales and MoE gating methods, the focus of this study is on foundational gradient estimation. Scaling and adapting our method for varied MoE paradigms presents both research and engineering challenges. We aim to explore these in future work, considering our current computational resource constraints.

---

### Official Review · Reviewer_4GtZ · 2023-11-02

**Soundness:** 3 good
**Presentation:** 3 good
**Contribution:** 2 fair
**Rating:** 5
**Confidence:** 4

**Summary:**

This paper attempts to address the backpropagation issue of Mixture of Experts (MoE). There is a growing interest in sparsely activated networks, specifically Mixture-of-Experts (MoE), which selectively activate parts of modules for specific inputs, leading to efficiency improvements. However, gradient estimation in MoE is challenging due to the non-differentiable nature of expert routing. Existing methods like Straight-Through (ST) estimators are not compatible with MoE, which leads to the neglect of gradient computation for routing during training, impacting convergence and model quality. The paper introduces SparseMixer, a novel approach that reconciles sparse MoE routing and backpropagation using numerical methods for ordinary differential equations. SparseMixer provides reliable gradient approximations even with a subset of experts activated and accelerates training convergence by up to two times. It also enables MoE models to consistently outperform dense models when used with Switch Transformer.

**Strengths:**

(1) This paper discusses an very important research question in MoE, the backpropoagation issue of routing function, which is easily be overlooked by researchers if not be pointed out specifically. This research question is timely and important.

(2)  The first order approximation used in this paper only requires the output of one expert, not sacrificing scalability.

(3) SparseMixer does not require hessian or other second-order derivatives, having negligible computation overheads.

**Weaknesses:**

(1) While SparseMixer achieves consistently improvement over the vanilla Switch Transformer, what I can see is the improvement is a bit marginal in Table1. Esp. as the number of experts increases, the performance gains become more marginal. My conjecture is that the main evaluation task in the paper, GLUE, is two simple to demonstrate the empirical benefits of SparseMixer. I would like to see more results on more challenging tasks, where the performance gains of S+S might be larger.

(2) The marginal performance improvement is also contradictory to the authors' motivation. If the neglect of the gradient computation for
routing is indeed crucial for MoE, we should see much significant improvements.

(3) What is the cost induced by SparseMixer? Table 3 does not provide enough information to justify the overheads. Can we see any time difference as the number of experts continues to increase?

**Questions:**

Please see the above Weaknesses.

---

> ### Author Response · Authors · 2023-11-23
> **Rebuttal by Authors**
>
> Thank you for your comprehensive feedback. We appreciate the opportunity to clarify and expand on the strengths of our approach, with further elaborations to be included in the final paper.
>
> **Reply to weakness 1 (experiment design and performance on GLUE):**
>
> In our submission, we adhered to the experiment design of the existing study, focusing on normal-scale problems for controllability and resource efficiency. For example, choosing ELECTRA-base as the pre-training task allows us to conduct experiments with multiple settings. As a comparison, typical ELECTRA-base model training takes 125k steps and typical BERT-base model training takes 1M steps.
>
> As to performance, we believe the improvement brought by SparseMixer is significant. Regarding training convergence, SparseMixer accelerates model training by up to two times. Regarding downstream performance, the best performance achieved by SparseMixer is 28.72 on WMT’14 EN-DE (note our architecture configuration is based on Transformer-base) and 88.31 on GLUE (note our architecture configuration is based on Electra-base).
>
> It's important to note that the improvements on GLUE are actually significant, as GLUE is a very competitive benchmark. Most leading methods on the GLUE benchmark (as in https://gluebenchmark.com/leaderboard) are based on the Electra-large setting and their improvements over the previous methods are usually within 0.2 absolute points. In our study, simply applying SparseMixer to Electra-base training without any pre-training hyper-parameter re-tuning, we achieve a consistent 0.5 absolute point improvements.
>
> Expanding our experiments to larger-scale problems or different MoE training paradigms is an exciting direction for future research. We acknowledge the combined research and engineering challenges this entails and look forward to exploring these possibilities as resources permit.
>
> **Reply to weakness 2 (Gradient Computation and Connection to Current Practice)**
> Thanks for the insightful suggestion! In our revised manuscript, we delve deeper into the relationship between our proposed first-order approximation and existing MoE training methodologies. Our analysis reveals that the common practice of ignoring certain gradient components is akin to reducing gradient scaling, a factor that adaptive optimizers like Adam can inherently adjust to. This insight not only validates our approach but also elucidates the underlying mechanics of current MoE training practices.
>
> To demonstrate this, please see the comparison between Switch and SparseMixer-ablation-2 (it fixes such additional bias by changing the sampling and avoiding the routing gradient downscale) in Figure 1. For more detailed discussion, please see Section 3.3 in the revised submission.
>
> **Reply to weakness 3 (Computational Overheads of SparseMixer)**
> SparseMixer has the same compute complexity and memory complexity with the current practice (dropping $\nabla_0$ completely). To further demonstrate this, we visualized the FLOPS ratio of Switch and SparseMixer in Figure 5 for $\\{2,4,8,16,32,64,128,256,512,1024,2048,4096,8192,16384\\}$. The computation overheads brought by SparseMixer only compose for less than $0.1\%$ of the training FLOPS.

---

### Meta-Review · Area_Chair_FBkG · 2023-12-19

**Metareview:**

The paper proposes a new gradient approximation mechanisms for sparse mixtures of experts, complementary to e.g. Switch transformers. The paper was appreciated by all reviewers but a few recurrent weaknesses have remained, in particular lack of comparison to better baselines and sloppy writing in section 3. I agree with this criticism regarding these points. This combined with the lack of any strong support from any of the reviewers makes me reject the paper. I do agree with the authors that the paper did already answer some of the points raised by reviewers (e.g. compute overhead), but I believe the misunderstandings may also arise from the convoluted presentation in Section 3, or presentation overall (e.g. Figure 1 in the early pages)

Summary of post-rebuttal phase:

The review of Reviewer *D3YK* listed weaknesses that were initially vague. I asked for clarifications before reviews were released, but these clarifications were only added later, in the discussion phase. The reviewer reacted stating:

> I find the method's ability to improve the performance to be marginal compared to the switch transformer baseline. Though, as a contribution I think the direction of sparse backprop in MoE is an interesting direction, I failed to appreciate the work as a whole due to its limited scope. Additionally, I don't identify myself to be an expert to pose enough critic on the theoretical underpinning of the current work (including section 3). However, I am also not in favor of discarding a work just because the results are not good enough to strongly back the current contribution. Thus, combined, I kept my original review at the fence. Though I did not explicitly mention (as suggested by you), to study the generalizability of this sparse gradient update across other transformer variants as well as other routing methods, I believe, it is quite implied to show validity of a method on at least more than one variant of architecture, which the authors have failed here. I intentionally kept that part a bit open ended as training within 10 days of rebuttal period is quite difficult and expected the authors to come up with something of their choice, however, they failed. [...] as an experimentalist I believe this work needs a revision._

Reviewer *yUXn* also reacted post-rebuttal,

> Although the authors took the whole rebuttal period to post the replies, I felt that the rebuttals and revision are quite hasty. For example:
▪️ The newly added results in Figure 1 are poorly discussed, the Figure is placed 3 pages before it is mentioned.
▪️ The contribution of omega scaling is poorly discussed. Given Figure 1, I could argue that in many cases, most of the contribution comes from the
omega scaling while the proposed method only contribute marginally.
▪️ The authors failed to report the total training time or explain why SparseMixer could achieve the same complexity despite introducing several computational steps.
▪️ The replies regarding to the extension to $k\geq 2$ or application on different architectures are not satisfactory. I expected the authors to outline some steps or challenges to extend to cases $k \geq 2$ and provide a minimal experiment/insight with a different architecture. However, none of such results are presented. Naming convention (minor): the name Sparse Mixer has been used in an existing work [A] for a different method. Although the approach is somewhat interesting, I believe the current draft still has many issues. Thus, I keep my original rating of this work and don't endorse the paper to be accepted in its current state. [A] Lee-Thorp, James, and Joshua Ainslie. "Sparse Mixers: Combining MoE and Mixing to build a more efficient BERT." arXiv preprint arXiv:2205.12399 (2022).




minor comment
- "To better understand the benefit introducing ω (as in Section 3.3)" -> wrong section

**Justification For Why Not Higher Score:**

The paper is below the borderline fence, as it failed to convince any of the reviewers. The contribution is fairly hacky, and discussion rushed. Experiments should be improved with more baseline methods, not just Switch Transformers, but any other more recent versions (e.g. S-BASE)

**Justification For Why Not Lower Score:**

NA

---

### Decision · Program_Chairs · 2024-01-16

Reject